# BRAF Inhibitors in Non-Small Cell Lung Cancer

**DOI:** 10.3390/cancers14194863

**Published:** 2022-10-05

**Authors:** Vincenzo Sforza, Giuliano Palumbo, Priscilla Cascetta, Guido Carillio, Anna Manzo, Agnese Montanino, Claudia Sandomenico, Raffaele Costanzo, Giovanna Esposito, Francesca Laudato, Simona Damiano, Cira Antonietta Forte, Giulia Frosini, Stefano Farese, Maria Carmela Piccirillo, Giacomo Pascarella, Nicola Normanno, Alessandro Morabito

**Affiliations:** 1Thoracic Medical Oncology, Istituto Nazionale Tumori, “Fondazione G. Pascale”—IRCCS, 80131 Napoli, Italy; 2Department of Medical Oncology, Institut Gustave Roussy, 114 Rue Edouard Vaillant, 94805 Villejuif, France; 3Department of Oncology and Hematology, Azienda Ospedaliera Pugliese-Ciaccio, 88100 Catanzaro, Italy; 4Oncology, San Giuseppe Moscati Hospital, 81031 Aversa, Italy; 5Clinical Trials Unit, Istituto Nazionale Tumori, “Fondazione G. Pascale”—IRCCS, 80131 Napoli, Italy; 6Scientific Directorate, Istituto Nazionale Tumori “Fondazione G. Pascale”—IRCCS, 80131 Napoli, Italy; 7Cellular Biology and Biotherapy, Istituto Nazionale Tumori, “Fondazione G. Pascale”—IRCCS, 80131 Napoli, Italy

**Keywords:** NSCLC, BRAF, MEK, dabrafenib, trametinib, encorafenib, binimetinib

## Abstract

**Simple Summary:**

The identification of BRAF mutations in 1.5–3.5% of NSCLC patients represents a step forward towards new targeted agents for tackling NSCLC. The positive results and favourable safety profile observed with the combination of dabrafenib and trametinib for the treatment of BRAF V600E metastatic NSCLC patients, led to the approval of these two agents in this setting. In this review, we will discuss the clinical, prognostic and therapeutic implications of BRAF mutations in NSCLC patients. Moreover, we will review MEKs’ functions and their role as a potential targeted treatment, both alone and combined with anti-BRAF therapies. Finally, we will discuss ongoing studies and future perspectives in this field. The clinical research on BRAF inhibitors in NSCLC is just beginning, and a number of relevant questions remain to be addressed regarding the role of other BRAF inhibitors in NSCLC, the possibility of facing resistance mechanisms to BRAF inhibitors, and the efficacy of other therapeutic strategies for BRAF-mutated NSCLC.

**Abstract:**

RAF family proteins are serine–threonine kinases that play a central role in the MAPK pathway which is involved in embryogenesis, cell differentiation, cell proliferation and death. Deregulation of this pathway is found in up to 30% of all human cancers and BRAF mutations can be identified in 1.5–3.5% of NSCLC patients. Following the positive results obtained through the combination of BRAF and MEK inhibitors in BRAF-mutant melanoma, the same combination was prospectively assessed in BRAF-mutant NSCLC. In cohort B of the BRF113928 trial, 57 pretreated NSCLC patients were treated with dabrafenib plus trametinib: an ORR of 68.4%, a disease control rate of 80.7%, a median PFS of 10.2 months and a median OS of 18.2 months were observed. Similar results were reported in the first-line setting (cohort C), with an ORR of 63.9%, a DCR of 75% and a median PFS and OS of 10.2 and 17.3 months, respectively. The combination was well tolerated: the main adverse events were pyrexia (64%), nausea (56%), diarrhoea (56%), fatigue (36%), oedema (36%) and vomiting (33%). These positive results led to the approval of the combination of dabrafenib and trametinib for the treatment of BRAF V600E metastatic NSCLC patients regardless of previous therapy. Ongoing research should better define the role of new generation RAF inhibitors for patients with acquired resistance, the activity of chemo-immunotherapy or the combination of TKIs with chemotherapy or with immunotherapy in patients with BRAF-mutated cancers.

## 1. Introduction

Lung cancer remains a significant cause of death worldwide, despite significant progress in recent years. The RAS-RAF-MEK-ERK (MAPK) pathway is amongst the main cascade which transduces extracellular signals into cellular responses. Under physiological circumstances, this pathway is involved in embryogenesis, cell differentiation, cell proliferation and death [1,2]. However, deregulation of this pathway is found in up to 30% of all human cancers and BRAF mutations can be identified in 1.5–3.5% of NSCLC patients. According to current international guidelines, patients with advanced NSCLC should be tested at least for EGFR and BRAF mutations and for ALK/ROS1 rearrangements to define a correct strategy of treatment [3]. Moreover, more recently, drugs targeting MET exon-14 skipping mutations and RET and NTRK fusions have become available, adding new relevant therapeutic options for NSCLC patients. Therefore, in this context, the identification of BRAF mutations represents a step forward towards new targeted agents for tackling NSCLC. In this paper, we will provide an updated review of clinical, prognostic and therapeutic implications of BRAF mutations in NSCLC patients, citing the more recent literature; this topic is particularly relevant in consideration of the rapid evolution of the knowledge in this field in the last few years. Moreover, we will deal with the role of MEKs as a potential targeted treatment, both alone and combined with anti-BRAF therapies. Finally, we will present the ongoing studies with BRAF inhibitors, and we will discuss future perspectives in this field, trying to address a number of different questions regarding the possibility of facing resistance mechanisms to BRAF inhibitors and the role of new strategies of treatment for patients with BRAF mutations. This information can be helpful for further developments in clinical studies.

## 2. MAPK Pathway and Its Alterations

RAF family proteins are serine–threonine kinases which play a central role in the MAPK pathway. Three different RAF isoforms can be found in mammals (ARAF, BRAF and CRAF) which are encoded by three different genes. Structurally, each RAF isoform is made up of three conserved regions (CR1, CR2 and CR3), containing the catalytic kinase domain (CR3), RAS binding domain (CR1) and regulatory domain (CR2). In addition to these constant regions, each protein also contains isoform-specific domains, which are probably responsible for different embryological functions observed for each RAF isoform [1,2]. Under physiological circumstances and regardless of different isoforms, RAF activation depends on the RAS-GTP binding state, which in turn is stimulated by growth factor receptors. Once activated, RAF isoforms create either homo- or heterodimers, eventually leading to the phosphorylation of MEK 1/2 and their downstream pathway (Figure 1) [4]. Among all of the RAF isoforms, BRAF is more frequently expressed in neural tissue and its deficiency has been correlated with abnormal neuroepithelial differentiation during embryogenesis. Despite being all activated by RAS-GTP, RAF isoforms show some differences in terms of their co-activators and regulators. As such, BRAF activation strictly depends on RAS, whereas BRAF itself may serve as a co-activating factor for CRAF with possible crosstalk between different isoforms at this level [1,2]. Furthermore, compared to CRAF, BRAF is a stronger activator of MEK 1/2, probably due to a higher affinity for this substrate [5]. MEK 1 and 2 proteins are ubiquitously expressed kinases also implied in the MAPK pathway, whose deficiency has been found to be lethal in mice. Structurally, both MEK 1 and 2 consist of three different regions: an N-terminal region implied in binding to its substrate, a core kinase domain and a C-terminal region. To date, ERKs are the only known substrate of MEKs, which in turn represent the only substrate of RAF family kinases. MEK activation requires RAF-mediated phosphorylation at two different serine residues. Consequentially, MEK proteins phosphorylate both the tyrosine and threonine residues of ERK 1/2 according to a double-step process, finally enhancing transcriptional factors implied in cellular growth, proliferation and evasion from apoptosis [6,7]. The MAPK pathway is frequently dysfunctional in many human cancers, the most common alterations being RAS and BRAF mutations (22% and 7% of all human tumours, respectively). On the other hand, MEK alterations and ARAF and CRAF mutations are extremely rare. Functionally, BRAF mutations have been dived into three different classes (class I, class II and class III), which differ in terms of RAS-dependency and activity of each catalytic domain. Class I BRAF mutations are constitutively functioning regardless of RAS signalling (RAS-independent) and have a higher kinase activity even in their monomeric status. Classically, BRAF V600 E/K/D mutants belong to this class. Class II BRAF mutations are also RAS-independent. Due to an intermediate kinase activity as monomers, however, class II proteins require dimer formation with other BRAF mutants (homodimers). This class includes some non-V600 point mutations such as K601E, L597Q and G469A, as well as BRAF chromosomal rearrangements (fusions/deletions). In contrast with the previous classes, class III BRAF mutations are RAS-dependent and are often associated with high RAS activity. Having an impaired kinase activity per se, these mutants need to form dimers with other wild-type CRAF isoforms (heterodimers) in order to be fully active. Class III mutants include D594 and G466 point mutations. BRAF class I and class II mutants are RAS-independent, and their function does not require any upstream pathway activation. Furthermore, given the inhibitory effects exerted by phosphorylated ERK to RAS proteins, class I and II BRAF mutations are usually characterized by low levels of RAS activity. These mutations are commonly known as “activator” and are usually mutually exclusive with other co-concurrent mutations. Conversely, class III BRAF mutants are known as “amplifier” since they strongly depend on upstream pathways. Indeed, class III proteins often occur with other mutations upstream in this pathway (e.g., RAS of NF1). To date, approved anti-BRAF drugs mainly inhibit class I BRAF mutations. Although rare (<1% of all tumours), MEK alterations are divided into three classes as well: RAF-independent, RAF-regulated and RAF-dependent alterations. RAF-independent MEK alterations are an effect of in-frame deletions which result in the loss of its regulatory domain. These mutations translate into a hyperactive MEK domain and do not occur with other mutations upstream. Instead, RAF-regulated and RAF-dependent MEK mutations display a certain kinase activity but still require RAF phosphorylation for perfect functioning. Anti-MEK molecules currently developed are allosteric inhibitors which block MEK in its inactive form, regardless of its mutational status. Since hyperactivation of the upstream pathway might reduce their inhibitory effect, MEK inhibitors usually have a low efficacy when given as monotherapy [8].

### 2.1. BRAF and MEK Alterations in NSCLC

In preclinical models of lung cancer, BRAF mutations seem to promote tumourigenesis at an early stage, particularly when other mutations co-occur [9]. In NSCLC patients, BRAF mutations are found in up to 4% of cases and are classically divided into V600E and non-V600E mutants, each of them representing roughly 50% of all cases [10,11]. Overall, BRAF mutations have been more frequently associated with a current or former smoking status and adenocarcinoma histology, although few other tumour types have been reported [12,13,14]. However, clinical and pathological characteristics of patients harbouring BRAF mutations may differ according to each functional class considered. Indeed, BRAF V600E cases have been associated with micropapillary growth patterns, while non-V600E may rather show various morphologies, including mucinous characteristics [15]. Furthermore, BRAF V600E mutations seem to be more predominant in females with no smoking history according to some retrospective studies, while non-V600 mutations are apparently related to current or former male smokers [16,17,18]. Differences among mutant classes could also concern metastatic pattern behaviour and prognostic outcomes. In a retrospective study evaluating 105 NSCLC patients, non-V600 patients had an up to four-fold higher rate of abdominal metastasis compared to V600 mutants (44% versus 12%, respectively), while the involvement of the pleura was less common in non-V600 patients (15% versus 21% for non-V600 and V600, respectively). Interestingly, no significant differences in terms of imaging features of the primary tumours among these functional classes have been observed [19]. Further retrospective data also showed that class I BRAF-mutant NSCLC patients had a significantly lower incidence of brain metastasis compared with other mutant classes (BRAF class I brain mets: 9%, class II: 29%, class III: 31%). Strikingly, class I patients had a significantly better overall survival compared with class II and III (class II vs. I: HR 2.50; 95% confidence interval (CI), 1.44–4.32; *p* < 0.001; and class III vs. I: HR 1.97; 95% CI, 1.09–3.56; *p*= 0.023), and a similar trend was observed when taking into account progression-free survival after first-line platinum-base chemotherapy (class II vs. class I: HR 1.80; 95% CI, 0.95–3.42; *p* = 0.069; class III vs. class I: HR 2.31; 95% CI, 1.04–5.15; *p* = 0.034) [20]. However, differences in survival for BRAF-mutated patients have not been further confirmed, suggesting that the real prognostic impact of BRAF alterations needs to be further elucidated [21]. Compared to other molecular subtypes of oncogene-addicted NSCLC, BRAF mutations may more frequently co-occur with other pathogenic alterations. By sequencing 392 tissue samples, BRAF mutations were found in 3.5% of all cases with 90% of those having concurrent mutations, mainly TP53, tyrosine kinase receptors (RTK) and STK11 mutations [22]. Comparable results were also obtained when analysing cfDNA obtained from BRAF-mutant NSCLC patients, the most frequently co-mutated genes being TP53 (57%), EGFR (26%), KRAS (15%) and NF1 (15%). Again, the prevalence of these concurrent mutations may vary according to each BRAF mutant class. Indeed, class III BRAF mutations are more likely associated with KRAS mutations than class I and II, whereas BRAF class II mutations might be more often associated with NF1 mutations than class I [11]. Although few cases of concomitant BRAF and EGFR mutations have been reported, BRAF mutations are generally considered to be mutually exclusive with other driver mutations [23,24]. Interestingly, BRAF mutations represent a known mechanism of both primary and acquired resistance to anti-EGFR TKIs, so far that patients harbouring EGFR and BRAF co-mutations progressed more rapidly upon EGFR inhibitors [25,26]. Overall, BRAF mutants seem to be more immunogenic than other molecular subtypes, with some variations displayed among different mutant classes. In a cohort of 39 BRAF-mutant NSCLC patients, PD-L1 expression ≥50% has been reported in 42% and 50% of BRAF V600E and non-V600E cases, respectively. These data being almost two-fold higher than the overall NSCLC population. Furthermore, a statistical but not a clinical difference in terms of PD-L1 expression favouring BRAF V600E mutants has been demonstrated (*p* = 0.05), whereas both these subgroups showed low/intermediate TMB and microsatellite-stable status [27]. Whether this more immunogenic status could translate into a higher response to immune checkpoint inhibitors needs to be further addressed, even though some initial promising data have been reported [27,28,29,30]. Finally, BRAF alterations other than mutations are anecdotal in NSCLC. By analysing 17128 tumour samples, BRAF fusions were identified in 42 cases (0.2%), the most frequent partner genes being AGK, DOCK4 and TRIM24. These cases were associated with low TMB, and some genetic co-alteration has been identified such as TP53 (67%), CDKN2A (31%), EGFR (29%) and CDKN2B (26%) [31]. To date, little is known about MEK alterations in NSCLC, mainly due to their low prevalence. In 2008, Marks et al. identified MEK1 mutations in 2 out of 207 NSCLC patients and both of these cases were not associated with other concurrent alterations [32]. A larger study also demonstrated that MEK mutations occurred in 36 among 6024 lung adenocarcinoma cases (0.6%), all of them being mutually exclusive with other oncogenic drivers and apparently associated with current smoking status [33]. Taken together, these data highlight that BRAF-positive NSCLC is rather a heterogeneous disease, whose main characteristics may strictly depend on single functional classes. Given the low prevalence of MEK mutations, no clinical implication of MEK alterations in NSCLC patients can be hypothesised.

### 2.2. BRAF and MEK Inhibitors in NSCLC

Since the identification of BRAF mutations among different types of cancers and the understanding of their pathological and clinical implications, several BRAF inhibitors (BRAFi) have been developed. Following the positive results obtained in BRAF V600E mutant patients with metastatic melanoma, the role of anti-BRAF drugs was investigated in other tumour types harbouring the same mutation [34,35,36]. Initial reports of anti-BRAF activity in NSCLC patients came from case reports attesting a dramatic response with vemurafenib in V600E BRAF-mutated NSCLC [37,38]. The retrospective multicentre EURAF cohort described the clinical course of patients treated outside of clinical trials with BRAFi [39]. A total of 34 patients with advanced stage NSCLC BRAF mutant (83% V600E) were treated with vemurafenib (*n* = 29), 5 of them were treated as first-line systemic treatment, dabrafenib (*n* = 9) or sorafenib (*n* = 1). In the sub-cohort of patients with BRAF V600E mutations receiving vemurafenib, 13 out of 25 (54%) achieved a disease response, with a disease-control rate (DCR) of 96%. One out of six patients with non-V600E mutations achieved a partial response with vemurafenib (harbouring G596V mutations), while the others (harbouring G466V, G469A, G469L, V600K and K601E mutations) did not respond to treatments. The non-V600E sub-cohort showed a poor outcome, with an overall survival (OS) with first-line therapy of 11.8 months versus 25.3 months for V600E patients. Irrespective of the specific treatment, the overall response rate (ORR) was 53%, the DCR was 85%, while the median progression-free survival (mPFS) and median overall survival (mOS) were 5.0 and 10.8 months, respectively [39]. The first prospective evidence of vemurafenib in NSCLC came from a histology-independent phase 2 basket trial of vemurafenib, 960 mg/oral twice daily, in BRAF V600 mutation-positive non-melanoma cancers (Table 1) [40]. An objective response was achieved in 8 (42%) out of 19 pretreated NSCLC patients, while a reduction in tumour size was described in 14 subjects. The PFS at 12 months was 23% (95% CI, 6–46), while mOS had not been reached but the preliminary rate was 66% (95% CI, 36–85). The final report of the expanded NSCLC cohort showed that 3 out of 8 previously untreated patients had a confirmed response while 5 had stable disease. In the subgroup of 54 previously treated patients, 20 had a confirmed response and 21 had stable disease. Overall, the objective response rate was 37.1% (95% CI, 25.2% to 50.3%), while in naïve patients it was 37.5% (95% CI, 8.5% to 75.5%) and 37.0% (24.3% to 51.3%) in pretreated patients. The median duration of response was 7.2 months (95% CI, 5.5–18.4), while the overall mPFS and mOS rates were 6.5 (95% CI, 5.2 to 9.0 months) and 15.4 months (95% CI, 9.6 to 22.8 months), respectively. The untreated cohort achieved a better outcome with an mPFS of 12.9 months (95% CI, 4.0: not evaluable (NE)) and an mOS (95% CI, 6.0–NE) instead of the previously treated cohort, where an mPFS of 6.1 months (95% CI, 5.1–8.3) and an mOS of 15.4 (95% CI, 8.2–22.6) were reported. Safety data were similar to ones obtained from studies of vemurafenib in cutaneous melanoma. All patients experienced at least one adverse event (AE), the most common of them were nausea (40% of patients), hyperkeratosis (34%), decreased appetite (32%), arthralgia (31%) and cutaneous squamous cell carcinoma (SCC) (26%) [41].

In the French Acsé programme, Mazieres et al. reported data of vemurafenib in 96 patients harbouring the BRAF V600E mutation and 15 with BRAF non-V600 mutations. In the BRAF non-V600 cohort, no objective response was observed suggesting the lack of activity of the single anti-BRAF agent in these alterations. In the BRAF V600 cohort, 43 out of 96 pretreated patients had objective responses (ORR 44.8%) with a median response duration of 6.4 months (95% CI, 5.1–7.3), an mPFS of 5.2 months (95% CI, 3.8–6.8) and an OS of 10 months (95% CI, 6.8–15.7). The most frequently reported treatment-related AEs of any grade were asthenia (56%), decreased appetite (46%), acneiform dermatitis (37%) and nausea and diarrhoea (35%) [42]. Treatment with vemurafenib has also provided some evidence of activity on brain metastases and meningeal carcinomatosis in BRAF V600E mutated lung cancer, although in retrospective series [46,47].

The first evidence of dabrafenib activity in BRAF V600E NSCLC was reported in one patient who achieved a response in a phase 1 trial mainly enrolling BRAF-mutant melanoma patients [48]. In the abovementioned EURAF cohort, nine patients were treated with dabrafenib and reported similar results to those observed with vemurafenib. Of note, a disease response was also noted in one patient who was receiving dabrafenib after progression to vemurafenib.

Evaluation of dabrafenib in patients with BRAF mutant stage IV NSCLC was assessed in an open-label, multicohort phase 2 trial (BRF113928) [43]. In cohort A, 78 subjects received oral dabrafenib, 150 mg twice daily, after one or more prior chemotherapy regimens for metastatic disease and 6 patients received dabrafenib as first-line treatment. Four out of six patients naïve to treatments achieved an objective response with dabrafenib, while an ORR of 33% (95% CI, 23–45) was reported in pretreated patients. The DCR was 58% (95% CI, 46–67).

In this trial, a better ORR (38%) and DCR (65%) were achieved by patients who received only one prior treatment regimen compared to the highly pretreated ones (ORR of 29% and DCR of 50%). Responses were also durable, with a median duration of response (DOR) of 9.6 months (95% CI, 5.4–15.2 months), leading to a median PFS of 5.5 months (95% CI, 3.4–7.3 months) and median OS of 12.7 months (95% CI, 7.3–16.9 months). All of the patients experienced one or more AEs such as pyrexia (36%), asthenia (30%), and hyperkeratosis (30%); however, the rates of serious AEs were low, conferring dabrafenib to a manageable safety profile [43].

However, despite BRAFi demonstrating clinical activity in patients with BRAF-mutant NSCLC, the majority of patients developed resistance to those drugs, in larger part leading to a reactivation of the MAPK pathway [49]. Data from in vitro experience demonstrated that the addition of a MEK inhibitor to BRAF inhibitors may overcome or delay the development of acquired resistance to BRAF inhibitors by blocking ERK signalling. The addition of MEK inhibitors may also prevent the paradoxical MAPK pathway activation, responsible for the development of cutaneous squamous cell carcinoma [50,51]. Following the results obtained through the combination of BRAF and MEK inhibitors in BRAF-mutant melanoma [52,53], the same combination was prospectively assessed in BRAF-mutant NSCLC.

In cohort B of the above-mentioned BRF113928 trial, the clinical activity and safety of dabrafenib (150 mg twice daily) plus oral trametinib (2 mg once daily) were investigated in 57 pretreated BRAF V600E-mutant metastatic NSCLC patients. A total of 38 (67%) out of 57 patients had received one prior treatment, and 19 (33%) had received two–three prior treatments [44]. In an updated 5-year survival analysis, Plachard et al. reported an ORR of 68.4% (95% CI, 54.8–80.1), a DCR of 80.7% (95% CI, 68–90), while a median PFS and median DoR were 10.2 (95% CI, 6.9–16.7 months) and 9.8 months (95% CI, 6.9–18.3 months), respectively. Survival data estimated a median OS of 18.2 months (95% CI, 14.3–28.6 months); 26% of the pretreated patients were still alive at 4 years and 19% at 5 years since the combination treatment initiation [45]. The BRF113928 trial also evaluated the combination therapy in 36 patients in the first-line setting (cohort C). At a median follow-up of 16.3 months, responses and control of the disease were similar when compared to those reported in pretreated patients, with an ORR of 63.9%, a DCR of 75% and a DoR of 10.2 months. The median PFS and OS were 10.8 and 17.3 months, respectively, with 34% of the treatment-naïve patients still alive at 4 years and 22% at 5 years from the treatment beginning. The combination was well tolerated, with a safety profile similar to that observed in patients with metastatic melanoma treated with the same combination. The main adverse event was pyrexia (56%) followed by gastrointestinal events such as nausea (51%), vomiting (41%), diarrhoea (37%) and decreased appetite (33%). Other relevant toxicities were dry skin (39%), peripheral oedema (38%) and a cough (31%). Positive results were also collected from four patients with brain metastases, all of them manifesting at least intracranial disease control with either monotherapy or a combination regimen [45].

## 3. Discussion

The positive results and favourable safety profile from the BRF113928 trial led to the approval of the combination of dabrafenib and trametinib for the treatment of BRAF V600E metastatic NSCLC patients, regardless of previous therapy [54,55]. However, the clinical research on BRAF inhibitors in NSCLC is just beginning and a number of relevant questions remain to be addressed regarding the role of other BRAF inhibitors in NSCLC, the possibility of facing resistance mechanisms to BRAF inhibitors, and the efficacy of other therapeutic strategies for BRAF-mutated NSCLC. For the first question, new BRAF inhibitors are under development in NSCLC and a number of clinical trials are currently ongoing to evaluate new combinations of BRAF V600E inhibitors for metastatic NSCLC patients (Table 2).

Encorafenib represents a new generation of BRAF inhibitors, showing a more prolonged pharmacodynamic activity than other drugs of the same family. The drug acts as an ATP-competitive RAF kinase inhibitor, decreasing ERK phosphorylation and downregulating cyclin D1 [56]. In 2018, the FDA approved the combination of encorafenib and binimetinib (an anti-MEK1/2 protein kinase inhibitor) for the treatment of patients with unresectable or metastatic melanoma with a BRAF V600E or V600K mutation. The approval was based on a phase 3 randomized, active-controlled, open-label, multicentre trial (COLUMBUS), that enrolled 577 BRAF-mutated patients with unresectable or metastatic melanoma [57]. In this trial, patients treated with a combination of encorafenib and binimetinib achieved a longer PFS than vemurafenib and encorafenib alone. In the combination arm, the main adverse reactions reported were fatigue, nausea, diarrhoea, vomiting, abdominal pain and arthralgia. Subsequently, the positive results from the BEACON CRC (colorectal cancer) trial led the FDA to approve, in 2020, the combination of encorafenib and cetuximab (an anti-EGFR monoclonal antibody) in the setting of pretreated BRAF V600E mutant patients with metastatic colorectal cancer [58]. Among 665 patients with metastatic CRC with progressive disease after one or two previous treatment regimens, better outcome, in terms of OS and ORR, were achieved via this combination rather than standard therapy. The most common adverse reactions (≥25%) with encorafenib and cetuximab were fatigue, nausea, diarrhoea, acneiform dermatitis, abdominal pain, decreased appetite, arthralgia and rash.

The ENCO-BRAF trial is a phase 2 trial for V600E mutated stage IV NSCLC patients with encorafenib 450 mg per day plus binimetinib 45 mg BID. There are two cohorts in this trial: the first one enrols naïve patients, and the second one enrols already treated patients. The primary endpoint is ORR. The estimated enrollment is 119 patients. Results are awaited for in March 2026 [59]. Another phase 2 trial is enrolling BRAF V600E mutated patients NSCLC with metastatic disease; V600K or V600D mutated patients are considered for enrollment. All patients receive encorafenib 450 mg per day plus binimetinib 45 mg BID; the primary endpoint is ORR. The estimated enrollment is 97 patients. Results are awaited for in March 2024 [60]. The Landscape 1011 study is a phase 1/2 umbrella trial involving metastatic NSCLC patients treated with a new anti-PD1 drug called Sasanlimab, which is unique is in its subcutaneous administration [61]. Patients with a BRAF V600E mutation who join the substudy A of the trial receive encorafenib and binimetinib plus Sasanlimab. The primary outcome measure is the durable objective response rate. The estimated enrollment is 375 participants. Results are awaited for in late 2024 [62]. The ROME trial is a phase 2 proof of concept trial involving patients with different cancers, including NSCLC. Patients should have completed at least one treatment but no more than two anti-cancer treatments. Patients are treated according to their newfound mutation; in particular, patients with BRAF V600E mutations are randomized to be treated with vemurafenib + cobimetinib or with SOC therapy according to guidelines. The primary endpoint is ORR. The estimated enrollment is 384 patients; results are awaited for in 2024 [63]. The B-FAST trial is a multi-cohort phase 2/3 trial that enrols naïve NSCLC patients who are metastatic or stage III and not eligible for a local treatment. Cohort E of the trial is devoted to BRAF V600-positive patients found through a liquid biopsy. In this cohort, patients receive vemurafenib at the dose of 960 mg BID on days 1–21, then at the dose of 720 BID on days 22–28, in combination with cobimetinib on days 1–21 and Atezolizumab every 4 weeks on day 29. The enrollment of this cohort is already completed. The primary endpoint is time in response (TIR). Results are awaited for in 2024 [64].

**Table 2 cancers-14-04863-t002:** Main ongoing studies with BRAF inhibitors in NSCLC.

Trial	Phase	Setting	Stage	Pts	Treatment	Primary Endpoints
**NCT04526782** **(ENCO-BRAF Trial)** **[59]**	Phase 2	Naïve or subsequent lines (two different arms)	Extensive stage	119	Encorafenib + Binimetinib	ORR
**NCT03915951** **[60]**	Phase 2	Any line of treatment	Extensive stage	97	Encorafenib + binimetinib	ORR
**NCT04585815** **(Landscape 1011 trial)** **[62]**	Phase 1/2	Any line of treatment	Extensive stage	375	Sasalnimab + encorafenib + binimetinib	Durable objective response rate
**NCT04591431** **(ROME trial)** **[63]**	Phase 2	Subsequent lines (no more than two treatments are allowed)	Extensive stage	384	Vemurafenib + cobimetinib vs. SOC *	ORR
**NCT03178552** **(BFAST Trial)** **[64]**	Phase 2/3	First line	Extensive stage	1000	Vemurafenib + Cobimetinib+ Atezolizumab	Time in response (TIR)

* SOC: standard of care.

Despite the activity of BRAF inhibitors for the treatment of BRAF-mutant NSCLC, most patients experience disease progression due to acquired resistance to targeted drugs. The mechanisms of resistance, mainly studied in patients with metastatic melanoma, have shown that secondary resistance to BRAF inhibitors can be due to the bypass of the MAPK pathway via activation of alternative signalling pathways or reactivation of ERK signalling through the MAPK pathway or other mechanisms [49]. Mechanisms involved may include BRAF splice variants or BRAF gene amplification, with increased levels of BRAF V600E homodimers, or secondary mutations in other genes leading to BRAF-independent reactivation of ERK signallings, such as NRAS/KRAS or MEK mutations [65]. In addition, Rudin et al. have reported that acquired K-ras G12D mutation might be another mechanism of secondary resistance to dabrafenib [66]. Third-generation RAF inhibitors have been recently investigated in preclinical studies and are currently studied in early-phase trials; they might have overcome several known mechanisms of resistance to first-generation RAF inhibitors [67,68]. In particular, the dubbed “paradox breakers” suppress mutant BRAF cells without activating the MAPK pathway in cells bearing upstream activation (i.e., PLX7904, PLX8394), with improved safety and more durable efficacy than first-generation RAF inhibitors. Among them, CCT196969 and CCT241161 are called pan-RAF inhibitors as they are able to block ARAF, BRAF and CRAF isoforms with high affinity.

Finally, a strategy of treatment for BRAF-mutated patients could be represented also through immunotherapy with CTLA-4 or PD-1/PDL-1 inhibitors, which have shown high activity in tumours with a high mutational burden [69,70]: BRAF mutations are often observed in smokers, whose tumours generally express a high mutational load. A recent retrospective analysis confirmed that the outcomes of BRAF-mutated patients undergoing immune checkpoint inhibitors were similar to that of unselected NSCLC [71].

## 4. Conclusions

In conclusion, the combination of dabrafenib and trametinib is currently the standard therapy for BRAF-mutated NSCLC. Ongoing research should better define the role of new generation RAF inhibitors for patients with acquired resistance as well as agents targeting non-V600E mutations. Several studies are also investigating the activity of chemo-immunotherapy or the combination of TKIs with chemotherapy or with immunotherapy in patients with BRAF-mutated cancers.

## Figures and Tables

**Figure 1 cancers-14-04863-f001:**
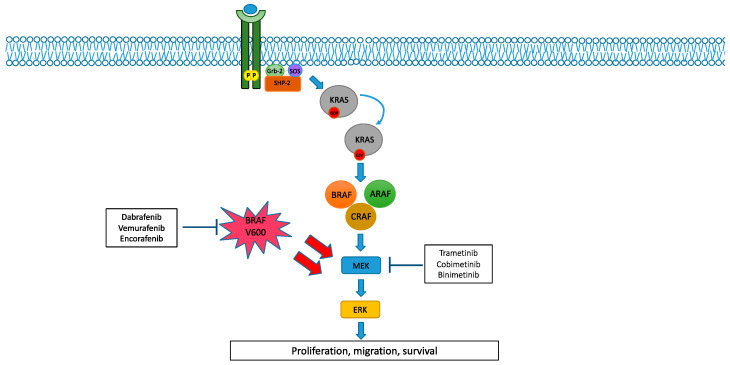
BRAF pathway and its inhibition. Under physiological conditions, BRAF activation strictly depends on RAS-GTP binding forms. Once activated, RAF isoforms create either homo- or heterodimers, eventually leading to the phosphorylation of MEK 1/2 and their downstream pathway. Conversely, BRAF class I mutations can directly activate downstream MEK 1/2 in an RAS-independent manner, resulting in deregulated proliferation and survival. BRAF inhibitors (dabrafenib, trametinib and encorafenib) combined with MEK inhibitors (trametinib, cobimetinib and binimetinib) eventually hinder downstream signalling.

**Table 1 cancers-14-04863-t001:** Clinical trials with BRAF and MEK inhibitors in NSCLC.

Study	Author	Setting	Pts	Treatment	Response Rate (%)	Disease Control Rate (%)	Progression-Free Survival (Months)	Overall Survival (Months)	Toxicity
**Phase 2** **(VE-BASKET study)**	Subbiah, 2019 [41]	Advanced solid tumours, Cohort NSCLC BRAF^V600^	62	Vemurafenib	37.1	-	6.5	15.4	Nausea, hyperkeratosis, decreased appetite, arthralgia, cutaneous SCC
**Phase 2** **(AcSé-BASKET study)**	Mazieres, 2020 [42]	Advanced solid tumours, Cohort NSCLC: BRAF^nonV600^ BRAF^V600^	115 15 100	Vemurafenib	0 44.8	- -	1.8 5.2	5.2 10	Asthenia, decreased appetite, acneiform dermatitis, nausea and diarrhoea
**Phase 2** **(BRF113928)**	Planchard 2016; 2022 [43,44,45]	BRAF^V600E^ advanced NSCLC: Cohort A pretreated Cohort B pretreated Cohort C naïve	171 78 57 36	Dabrafenib Dabrafenib+ Trametinib Dabrafenib+ Trametinib	33 68.4 63.9	58 80.7 75	5.5 10.2 10.8	12.7 18.2 17.3	Pyrexia, asthenia, hyperkeratosis, decreased appetite, nausea

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
