# Peer review of "BRAF Inhibitors in Non-Small Cell Lung Cancer"

_cancers, 2022, doi:10.3390/cancers14194863_

Round 1

Reviewer 1 Report

“BRAF inhibitors in Non Small Cell Lung Cancer” by Sforza et al. is a very well-written manuscript with mainly emphasizing on “BRAF and MEK alterations in NSCLC” and “BRAF and MEK inhibitors in NSCLC”. The information presented appear to be well collected organized. However, as a manuscript there appear to be some novelty issues as several manuscripts are already published with same theme. Following are some of them-

Anguera G, Majem M. BRAF inhibitors in metastatic non-small cell lung cancer. J Thorac Dis. 2018 Feb;10(2):589-592. doi: 10.21037/jtd.2018.01.129. PMID: 29607117; PMCID: PMC5864601.

Khunger A, Khunger M, Velcheti V. Dabrafenib in combination with trametinib in the treatment of patients with BRAF V600-positive advanced or metastatic non-small cell lung cancer: clinical evidence and experience. Ther Adv Respir Dis. 2018 Jan-Dec;12:1753466618767611. doi: 10.1177/1753466618767611. PMID: 29595366; PMCID: PMC5941661.

Odogwu L, Mathieu L, Blumenthal G, Larkins E, Goldberg KB, Griffin N, Bijwaard K, Lee EY, Philip R, Jiang X, Rodriguez L, McKee AE, Keegan P, Pazdur R. FDA Approval Summary: Dabrafenib and Trametinib for the Treatment of Metastatic Non-Small Cell Lung Cancers Harboring BRAF V600E Mutations. Oncologist. 2018 Jun;23(6):740-745. doi: 10.1634/theoncologist.2017-0642. Epub 2018 Feb 7. PMID: 29438093; PMCID: PMC6067947.

Baik CS, Myall NJ, Wakelee HA. Targeting BRAF-Mutant Non-Small Cell Lung Cancer: From Molecular Profiling to Rationally Designed Therapy. Oncologist. 2017 Jul;22(7):786-796. doi: 10.1634/theoncologist.2016-0458. Epub 2017 May 9. PMID: 28487464; PMCID: PMC5507646.

Sánchez-Torres JM, Viteri S, Molina MA, Rosell R. BRAF mutant non-small cell lung cancer and treatment with BRAF inhibitors. Transl Lung Cancer Res. 2013 Jun;2(3):244-50. doi: 10.3978/j.issn.2218-6751.2013.04.01. PMID: 25806238; PMCID: PMC4367599.

Alvarez JGB, Otterson GA. Agents to treat BRAF-mutant lung cancer. Drugs Context. 2019 Mar 13;8:212566. doi: 10.7573/dic.212566. PMID: 30899313; PMCID: PMC6419923.

Authors need to clarify how the current manuscript is different from previous publications in terms of novelty/updated information.

Other comments

·         1. Introduction: This part is too brief to formulate the question why this review is needed. A thorough background should be presented to support the rationale behind this review.

·         Page 4, line 195: “V6600E BRAF” should be “V600E BRAF”.

Author Response

A) Authors need to clarify how the current manuscript is different from previous publications in terms of novelty/updated information

Response: we thank the reviewer for her/his very positive comments and for all the suggestions. We specified in the end of the introduction how our review is different from previous publications on the same theme, adding the following paragraph: "In this paper, we will provide an updated review on clinical, prognostic and therapeutic implications of BRAF mutations in NSCLC patients, citing the more recent literature: this topic is particularly relevant in consideration of the rapid evolving of the knowledge in this field in the last few years. Moreover, we will deal with the role of MEK kinases as a potential targeted treatment, both alone or combined with anti-BRAF therapies. Finally, we will present the ongoing studies with BRAF inhibitors and we will discuss about future perspectives in this field, trying to address a number of different questions regarding the possibility to face resistance mechanisms to BRAF inhibitors and the role of new strategies of treatment for patients with BRAF mutations: this information can be helpful for further developments in clinical studies"

B) Introduction: This part is too brief to formulate the question why this review is needed. A thorough background should be presented to support the rationale behind this review.  

Response: we added in the introduction a background to support the rationale behind this review and to explain the importance of this topic. "According to current international guidelines, patients with advanced NSCLC should be tested at least for EGFR and BRAF mutations and for ALK/ROS1 rearrangements to define a correct strategy of treatment. Moreover, more recently, drugs targeting MET exon-14 skipping mutations, RET and NTRK fusions have become available, adding new relevant therapeutic options for NSCLC patients. Therefore, in this context, identification of BRAF mutations represents a step forward towards new targeted agents for tackling NSCLC."

C: Page 4, line 195: “V6600E BRAF” should be “V600E BRAF”.

Response: Corrected as suggested

Reviewer 2 Report

The article is very comprehensive and holds a good flow to it. It would be good to have some pictorial representations of the mechanism of action of the drugs and the multiple pathways as it will ease the reading process.

Author Response

A: The article is very comprehensive and holds a good flow to it. It would be good to have some pictorial representations of the mechanism of action of the drugs and the multiple pathways as it will ease the reading process.

Response: we thank the reviewer for the positive comments. We added a Figure 1 (on page 2, paragraph 2) to better explain the mechanism of action of the drugs and the multiple pathways with this legend to figure 1: “BRAF pathway and its inhibition. Under physiological conditions, BRAF activation strictly depends RAS-GTP binding forms.  Once activated, RAF isoforms create either homo or hetero dimers, eventually leading to phosphorylation of MEK 1/ 2 kinases and its downstream pathway. Conversely, BRAF class I mutations can directly activate downstream MEK 1/2 kinanes in a RAS- independent manner, resulting in deregulated proliferation and survival. BRAF inhibitors (dabrafenib, trametinib, encorafenib) combined with MEK inhibitors (trametinib, cobimetinib, binimetinib) eventually hinder downstream signalling.”    

Reviewer 3 Report

This article reviews BRAF inhibitors on non-small cell lung cancer (NSCLC). The authors described the effects of the combination of dabrafenib and trametinib for discussed BRAF mutations for therapeutic implications and other related treatment research. Since therapeutics for NSCLC still have room for improvement, the manuscript is valuable since it contains helpful information for further developments in clinical studies. Moreover, the manuscript cites more recent literature about the case studies. I recommend this manuscript to publish in Cancer with the following minor revision.

  • There are some occasional grammatical problems/typos within the text. You may need to ask someone fluent in English to enhance readability. For example,
    • Line 113 “and to not occur”  do not occur
    • Line 367 “may overcame”  might have overcome
  • Some sentences, for example, the last sentence in the conclusion, are too long, wordy, and hard to follow.
  • Many places are double-spaced in the middle of the sentence.
  • Add reference numbers in Tables 1 and 2, so the reader can easily check the references.

Author Response

A: This article reviews BRAF inhibitors on non-small cell lung cancer (NSCLC). The authors described the effects of the combination of dabrafenib and trametinib for discussed BRAF mutations for therapeutic implications and other related treatment research. Since therapeutics for NSCLC still have room for improvement, the manuscript is valuable since it contains helpful information for further developments in clinical studies. Moreover, the manuscript cites more recent literature about the case studies. I recommend this manuscript to publish in Cancer with the following minor revision.

Response: We thank the reviewer for the global positive comments and for the suggestions.

B: There are some occasional grammatical problems/typos within the text. You may need to ask someone fluent in English to enhance readability. For example: Line 113 “and to not occur” à do not occur; Line 367 “may overcame” à might have overcome. Some sentences, for example, the last sentence in the conclusion, are too long, wordy, and hard to follow. Many places are double-spaced in the middle of the sentence.

Response: we corrected the grammatical problems/typos within the text and we rewrote some sentences

C: Add reference numbers in Tables 1 and 2, so the reader can easily check the references.

Response: we added reference as suggested

Round 2

Reviewer 1 Report

I belive the changes have been incorporated by the authors.